# Effects of Multimodal Physical and Cognitive Fitness Training on Sustaining Mental Health and Job Readiness in a Military Cohort

Paul Taylor [1,*], Frederick Rohan Walker [2,3], Andrew Heathcote [1] and Eugene Aidman [3,4]

[1] School of Psychological Sciences, The University of Newcastle, Callaghan, NSW 2308, Australia; ajheathcote@gmail.com
[2] Centre for Advanced Training Systems, The University of Newcastle, Callaghan, NSW 2308, Australia; rohan.walker@newcastle.edu.au
[3] School of Biomedical Sciences & Pharmacy, The University of Newcastle, Callaghan, NSW 2308, Australia; eugene.aidman@defence.gov.au
[4] Division of Human and Decision Sciences, Defence Science & Technology Group, Edinburgh, SA 5111, Australia
[*] Correspondence: paul.taylor10@uon.edu.au; Tel.: +61-431386587

**Abstract:** Drawing on the emerging area of workplace sustainability, this study sought to measure the effects of multimodal physical and cognitive fitness training on sustaining mental health and job readiness via impacts on subjective burnout, mental wellbeing, and resilience in a military cohort. Volunteer participants were block randomised into either a standard 4-week resilient mind program (RMP) intervention or an RMP combined with self-paced functional imagery practice (RMP+FI). Self-reported burnout, mental wellbeing, and resilience were measured at baseline and at the end of the 4-week intervention using the Maslach Burnout Inventory-General Survey (MBI-GS), Brief Resilience Scale (BRS), and the World Health Organization's WHO-5 Well-Being Index (WHO-5), respectively. A total of 78 participants were enrolled in the study and 72 (92%) completed the program. Repeated measures ANOVAs showed significant effects of the RMP intervention, with both the RMP and RMP+FI groups reporting improved resilience (F(1, 70) = 13.08, $p < 0.001$, partial $\omega^2$ = 0.00086) and mental wellbeing (F(1, 70) = 41.86, $p < 0.001$, partial $\omega^2$ = 0.36). Both groups also reported improved burnout markers for professional efficacy (F(1, 70) = 6.25, $p < 0.002$, partial $\omega^2$ = 0.02), as well as reduced emotional exhaustion (F(1, 70) = 31.84, $p < 0.001$, partial $\omega^2$ = 0.02) and job cynicism (F(1, 70) = 8.80, $p < 0.005$, partial $\omega^2$ = 0.005). The FI practice produced no significant improvement in the RMP-only condition. Our results support the efficacy of RMP intervention in reducing burnout symptoms and improving self-reported mental wellbeing and resilience in a cohort of serving Navy aviators.

**Keywords:** mental health; sustainability; resilience; wellbeing; psychology

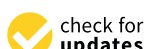



## 1. Introduction

Burnout is a psychological syndrome emerging in response to chronic job stress [1], recognised as an occupational hazard in a range of workplace settings, including corporations, first responder professions, and the military [2,3]. Burnout is associated with mood disturbances and significant deficits in cognitive functioning, including memory and attention, resulting in impaired performance and workplace sustainability [4].

Chronic stress at work is known to contribute to both burnout and the development of mood disorders, including depression, anxiety, and post-traumatic stress [5,6]. The psychology of sustainability and sustainable development calls for the achievement of sustainable wellbeing from a primary prevention point of view [7]. However, a major challenge for first responder and military professions is that significant levels of stress are unavoidable in their line of work, which often involves a combination of uncertainty, volatility, and complexity [8,9].

Recent military conflicts appear to have exacerbated the scale of stress-related injuries and sustainable working. In the 6 years from 2012 to 2018, medical discharges from the British Army for mental ill health, including post-traumatic stress disorder (PTSD), have tripled [8]; between 2009 and 2019, the US Army discharged approximately 22000 soldiers (out of a total serving population of around 480,000) for mental health problems following service in Iraq or Afghanistan [9]; and in April 2021, the Australian government announced the establishment of a Royal Commission to investigate the high suicide rates and incidence of mental health dysfunction amongst Australian military and veteran communities [10].

Given the importance of unimpaired cognitive function for sustainable working in military and first responder workforces, there is growing demand for strategies to prevent stress-related injury such as burnout. The research into burnout prevention has largely focused on the development of resilience training programs [2]. Definitions of resilience range from 'a stable equilibrium in spite of trauma' [3,11] to 'the ability to maintain normal psychological and physiological functioning in the presence of high stress and trauma' [12], to 'the capacity to overcome the negative effects of setbacks and associated stress on military performance and combat effectiveness' within military contexts [13]. Several resilience training programs developed to date include the US Comprehensive Soldier Fitness Program [14], the Canadian Police's Road to Mental Readiness [15], and the UK military's Mental Resilience Training program [16].

Evaluations of existing resilience programs have reported mixed results. A systematic review of thirty-three resilience training studies across twelve high-risk occupations in eight countries (including eleven with military cohorts) reported improved wellbeing outcomes and concluded that resilience training is generally effective for people in high-risk occupations [17]. However, the authors noted that the high level of reported success could be due to publication bias and that the results may not be generalisable beyond study populations [17]. Additionally, only nine of these studies measured burnout/stress, mood, or resilience as outcome measures. A 2018 review of Mental Resilience Training (MRT) developed and implemented within the British Army supported the roll-out based on an increased usage of psychological skills by participants, but did not include a validated assessment of resilience [16]. More recently, a randomised controlled trial (RCT) conducted with UK military recruits assessed the impact of a resilience-based intervention on mental health measures of participants and found no specific benefits of the program, and the authors noted that time and resources would not be well spent implementing such interventions without establishing their efficacy [18]. Additionally, the US military modelled their Comprehensive Soldier Fitness Program on resilience training for school-aged children, and the program has been criticised for insufficient evidence of its effectiveness before its large-scale roll-out to soldiers [14].

Nearly all resilience interventions mentioned above have utilised purely psychological modes of delivery. With the physical component missing, this may have been a limitation. A recent international military physiology roundtable suggested that resilience is a complex phenomenon influenced by physiological, psychological, and social factors [6], and recommended a combined 'psychophysiological' approach to military resilience training. Similar integrated approaches have been proposed, e.g., the Cognitive Fitness Framework (CF2) [19]. Growing evidence suggests that exercise can be beneficial for mood disorders [20]; breathwork can reduce stress—both at self-report and biomarker levels [21]—and cold showers have been shown to reduce absenteeism and sickness [22]. There are also some indications that the positive impacts of exposure to such stressors may cause cross-adaptation and cross-tolerance, whereby adaptation to one stressor develops cross-tolerance to resist the adverse effects of another type of stressor [23].

This study sought to examine the efficacy of a 4-week psychophysiological resilience training intervention—the Resilient Mind Program (RMP)—in improving mental wellbeing and resilience and reducing the risk of burnout in a cohort of serving military aviators. The RMP has previously been shown to be effective in improving self-reported wellbeing and resilience in corporate workplaces, including banking, insurance, and government

sectors [24]. The current study also examined the additional effect of functional imagery practice (FI) on the outcomes of RMP. FI is an imagery-based behavioural intervention designed to help people to enhance their motivation and sustain their behaviour using mental imagery. It typically combines person-centred counselling, such as motivational interviewing, with tailored imagery exercises to elicit strong emotional responses and strengthen motivation for change [25,26]. FI has been shown to enhance grit in soccer players [25], assist in weight loss [26], and reduce alcohol cravings [27].

## 2. Method

### 2.1. Research Design

A 2 × 2 repeated measures design was employed, with a pre/post-assessment of outcome measures as the within-participants factor and two training conditions (detailed in the Procedure subsection below) as the between-participant factor.

### 2.2. Participants

Seventy-eight Navy aviators (65 males, 11 females, mean age 35.9 years, SD = 8.9 years), from the Royal Australian Navy Fleet Air Arm, volunteered to participate between January and February 2020. Study eligibility required participants to (1) have active-duty status; (2) commit to study participation for 4 consecutive weeks, including a 3 h instructor-led education session; and (3) be free from any medical condition that would inhibit their ability to exercise. Briefly, 78 participants were enrolled in the study and 72 completed it. Only the data for those who completed the study were analysed.. Reasons for the dropouts were (1) voluntary withdrawal, (2) absent from work on the day of retesting, and (3) duty reassignment.

The study's protocol was approved by the Defence Science and Technology Group's Low-Risk Ethics Research Panel (LREP protocol LD 11-20) and was endorsed by the participants' chain of command. Participants were recruited via a naval-base-approved email, provided informed consent prior to commencing their participation in the study, and were informed that they were free to withdraw at any time without detriment to their careers.

### 2.3. Materials

Outcome measures were assessed using a battery of self-report questionnaires administered before and immediately after the 4-week intervention.

Mental wellbeing was measured using the World Health Organization's Wellbeing Index (WHO-5), which assesses mental wellbeing over the last 2 weeks using 5 items, such as 'I have felt cheerful and in good spirits'. Participants respond to each item on a 5-point Likert scale, ranging from 0 (at no time) to 5 (all of the time). Raw scores range from 0 to 25, with 0 representing the worst possible and 25 representing the best possible quality of life. The WHO-5 has been found to have adequate validity in screening for depression and in measuring outcomes in clinical trials, and item response theory analyses (IRT) in studies with the youth and elderly indicate that the measure has good construct validity as a unidimensional measure of wellbeing in these populations [28].

Psychological resilience was assessed using the Brief Resilience Scale (BRS) [29]. It contains 6 items, such as 'It is hard for me to snap back when something bad happens'. Participants respond to each item on a 5-point Likert scale, ranging from 1 (strongly disagree) to 5 (strongly agree). Half of the questions are reverse coded. Scores fed back to the participants ranged from 1 to 5, with scores below 3.00 indicating low resilience, scores above 4.30 indicating high resilience, and scores in between indicating average resilience. The BRS has been shown to have good reliability and validity estimates [30].

Burnout was assessed using the Maslach Burnout Inventory-General Survey (MBI-GS) which measures respondents' relationship with their work on a continuum from engagement to burnout [31]. The MBI-GS contains 16 items on a 7-point Likert scale ranging from 0 (never) to 6 (every day). The instrument measures 3 dimensions of burnout:

Exhaustion (5 items, such as 'I feel emotionally drained from my work'), Cynicism (5 items, such as 'I have become more cynical about whether my work contributes anything'), and Professional Efficacy (6 items, such as 'I can effectively solve the problems that arise in my work'). High scores on Exhaustion and Cynicism and low scores on Professional Efficacy equate to a high degree of burnout. The three-factor structure of the MBI-GS was established by conducting a confirmatory factor analysis using LISREL in several studies, and factor loadings can be accessed in the Maslach Burnout Inventory Manual, Fourth Edition. The reliability of the three MBI scales exceeds the recommended levels for research instruments, with an analysis of 84 studies finding reliability estimates for the Emotional Exhaustion scale to be in the high 0.8 s and the other 2 scales to be in the mid 0.7 s [31].

### 2.4. Procedure

In a block-randomised design, two work teams (who worked together on shifts, known as 'watches') of 40 and 38 participants were randomly allocated into one of two conditions:

(i)   Standard Resilient Mind Program: three hours of face-to-face delivery plus web application for 4 weeks (RMP, *n* = 38);

(ii)  Standard RMP + functional imagery practice: three hours of face-to-face delivery plus web application and self-paced FI practice for 4 weeks (RMP + FI, *n* = 40).

The RMP [24] is a combined physical and psychological intervention with 3 h of face-to-face psychoeducation augmented with a 4-week program via a web application that is designed to help participants form healthy habits and interact with each other. The psychoeducation draws on acceptance commitment therapy (ACT), mindfulness techniques, and cognitive reframing techniques, as well as physical 'rituals' including exercise, cold showers, rhythmic breathing, and sleep hygiene. The web application contains educational videos, workouts, guided breathing, and meditation sessions, a 'Ritual Board' for logging suggested actions, leaderboards for gamification, and a social feed (Figure 1). There is also a feature dedicated to FI practice, which was available to RMP+FI group.

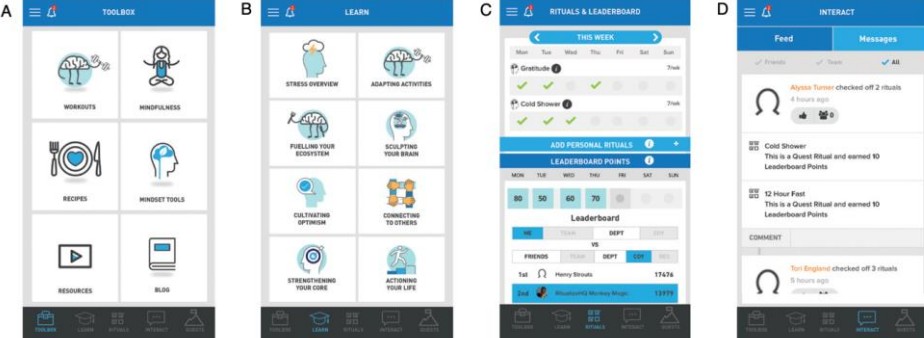

**Figure 1.** Screenshots of the Resilient Mind Program web application. (**A**) Toolbox with resources such as guided bodyweight workouts, guided meditation, breathing sessions, a range of healthy recipes, various mindset exercises, additional resources such as suggested books for reading and videos, and a range of video-based user guides for the app. (**B**) Learn section with a range of educational videos which participants could view at their leisure. (**C**) Ritual Board to track suggested habits and a points leaderboard. Participants earned points for ticking off rituals; completing workouts, guided meditation, and breathing sessions; and posting on the social feed. (**D**) Interact tab containing a social feed of user activities and posts. Users had the ability to make supported comments called nudges and earn points for doing so.

Participants had 4-week access to the web application and were encouraged to perform guided workouts, breathing and meditation sessions, and a number of key habits (rituals), which were pre-loaded onto a 'Ritual Board' for them to tick off once completed. Participants earned leaderboard points for all of these activities and had the option of setting reminders for 'rituals', delivered in the form of a pop-up notification on their phone.

Engagement with the platform was voluntary, with engagement time varying from a few minutes a week for those mainly using the Ritual Board, to an hour or more per week for those highly engaged with educational videos, breathing and meditation sessions, and workouts.

*2.5. Analysis*

All data analyses were conducted using R statistical functions and RStudio packages for data visualisation. Intervention effects were examined using linear mixed model ANOVAs [32], assuming a Gaussian error function estimating the size and significance of pre- to post-changes on each outcome measure (WHO-5, BRS, and the three MBI-GS subscales) from baseline to immediately after the 4 weeks [33]. Based on meta-analytic evidence from randomised control studies of the effects of similar programs on stress [34], an effect size of d = 0.4 on training-induced change can be considered to be meaningful and practically significant. This effect size is equivalent to f = 0.25 for ANOVA power calculations. To reliably detect effects of this size and to improve the generalisability of findings, with two groups and two repeat measurements in our design, G*Power calculations return a minimal sample size of 27 participants per group [35]. To allow for sample attrition, 40 and 38 participants were recruited to the two groups.

## 3. Results

A total of 78 participants were enrolled in the study and 72 (92%) completed the study. Data analysis was only performed on the data of those who completed the study.

*Effects of the RMP and FI Practice Intervention*

Repeated measures ANOVAs revealed the significant main effects of the RMP intervention, with both groups reporting improved mental wellbeing ($F_{(1, 70)} = 41.86$, $p < 0.001$, partial $\omega^2 = 0.36$) (Figure 2A) and resilience ($F_{(1, 70)} = 13.08$, $p < 0.001$, $\omega^2 = 0.00086$) (Figure 2B). Both groups also reported improved scores on the burnout subscale of professional efficacy ($F_{(1, 70)} = 6.25$, $p < 0.002$, partial $\omega^2 = 0.02$) (Figure 3A), as well as reduced burnout scores on the subscales of emotional exhaustion ($F_{(1, 70)} = 31.84$, $p < 0.001$, partial $\omega^2 = 0.02$) (Figure 3B) and job cynicism ($F_{(1, 70)} = 8.80$, $p < 0.005$, partial $\omega^2 = 0.005$) (Figure 3C).

No main effect of FI practice or interaction between time (pre- to post-training) and group (RMP+FI versus RMP only) was observed for any of the outcome measures (all Fs < 1.0, $p > 0.05$), indicating that the additional FI practice did not produce a significant improvement in the RMP-only condition (Figures 2 and 3).

Participant engagement, measured as leaderboard points scored for interacting with the various aspects of the mobile application, was similar in both groups (1789 points on average for participants in the RMP group and 1668 points on average in the RMP + FI group). The time spent on the app showed no significant differences between groups.

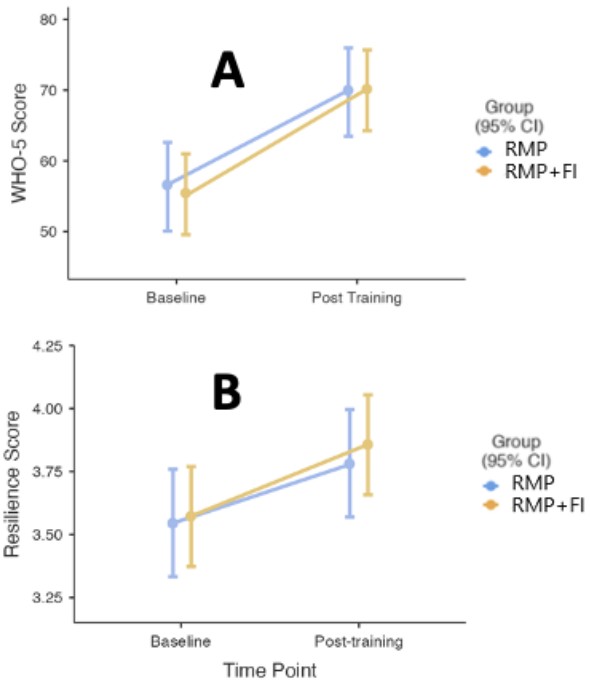

**Figure 2.** Pre- to post-training change in mental wellbeing (**A**) and resilience (**B**) scores for the RMP training group (blue line) and the RMP plus FI training group (orange line).

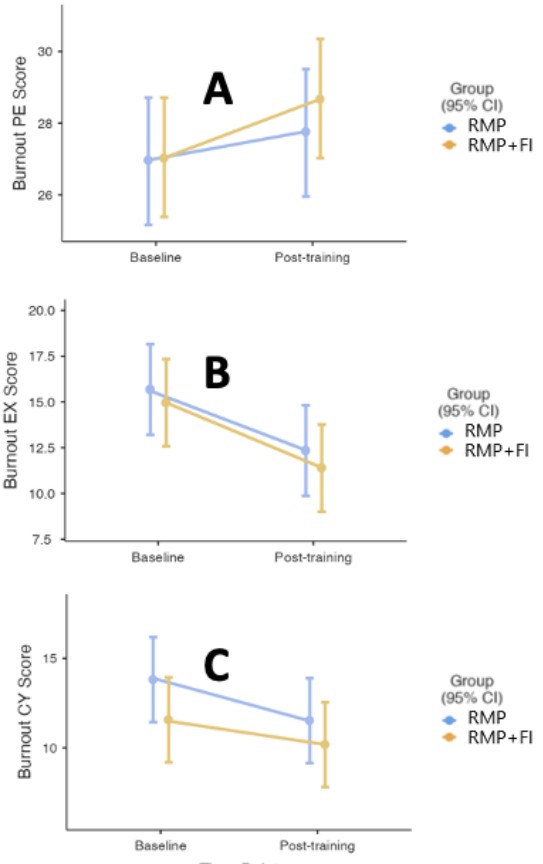

**Figure 3.** Pre- to post-training change in burnout scores for the RMP training group (blue line) and the RMP plus FI training group (orange line): professional efficacy (**A**), emotional exhaustion (**B**), and job cynicism (**C**).



## 4. Discussion

In this study we examined the effect of a combined physical and psychological intervention (RMP) on multiple self-report markers of mood, resilience, burnout, and overall sustainable working. We also sought to ascertain whether additional functional imagery practice could modulate the effect of the RMP program on these measures. The main finding was that the RMP produced significant improvements in self-reported mood, resilience, and burnout symptoms, compared to baseline, in both the RMP and RMP + FI groups. However, the magnitude of these improvements did not differ between the two groups, indicating that FI practice, as implemented in our study, did not amplify the training effects of RMP itself.

The main contribution of this study is that observed improvements in mood, resilience, and burnout demonstrate that the brief training program is quite effective at enhancing workplace sustainability, at least at the level of self-report measures. Although general improvements have been noted in a systematic review of resilience training interventions [17], to our knowledge this is one of the first demonstrations that a program of this nature, with serving military personnel, involving a relatively modest face-to-face training exposure, in combination with a phone-app-supported self-directed practice, can induce positive changes in self-reported measures of mood, resilience, and burnout, and therefore workplace sustainability. The results of our study contrast a resilience-based RCT for UK military recruits, which found no evidence of any specific benefit of the program to participants, although that study used a different measure of mental wellbeing (the 12-item General Health Questionnaire) [18].

The level of engagement in the RMP group was high, with 72 out of 78 participants completing the training, which is much higher than the internet-based Resilience at Work (RAW) mindfulness program, which was piloted in first responders, with 33% of participants completing that program and reporting nonsignificant improvements in resilience [36]. Possible reasons for the increased engagement and changes in resilience could be the addition of 3 h of face-to-face in the RMP versus online only in the RAW program, or the combined physical and psychological training in the RMP versus psychological-only training in the RAW program, or both. Indeed, a recent report of an international military physiology roundtable recommended that, due to the complex nature of resilience in high-risk settings, multimodal approaches to developing resilience are preferable [13]. The effectiveness of a multimodal approach is underscored by our study and a recent comprehensive multimodal intervention with active-duty US Airmen, which showed improvements in both physical fitness and a range of cognitive performance measures; however, no measures of mood or resilience were included in this study [37].

Our observation that the RMP+FI group did not show any additional benefit over the RMP group was contrary to expectations, given the previously demonstrated benefits of FI practice [24]. There are, nevertheless, several methodological factors that may have contributed to the lack of any observable FI impact. First, and most importantly, our induction of FI practice may have been suboptimal—participants in our study were given a brief guide to FI practice and instructed to use the feature in the app, whereas in other FI interventions, participants are guided through a series of mental imagery exercises and trained to practice imagery at home [24–26]. Second, we did not have the functionality to track the number of FI practice sessions that the users completed; so, there is the possibility that uptake was low and supervised sessions may have elicited greater effects. Third, the other elements of the program may have combined to reach a benefits ceiling. Given these limitations, we conclude that the further investigation of potential additional benefits of a more comprehensive, supervised FI practice may be warranted in other interventions.

There are several potential limitations to the study. First, and most notably, was the lack of a control group. Second, was the relatively small sample size, which, although sufficiently powered, was limited in nature due to the need to assess the in-principal feasibility of the program in a serving military cohort of limited size before any wider roll-out, given previous criticisms of military training programs that have been rolled out

on a large scale without any evidence of effectiveness [14,18]. Third, was the use of self-reported measures of mood, resilience, and burnout, although all of the measurement scales used were well validated, and the burnout measure is a commonly used scale to assess nonclinical burnout [1]. Fourth, we did not analyse the effect of the individual components of the multimodal program and further research is required to assess the contributions of the various physical and psychological elements. Fifth was the absence of a long-term follow-up to assess ongoing effects.

Although tempered by the study's limitations, the results for sustainable working are promising in terms of the efficacy of a multimodal approach to cognitive fitness training that combines physical and psychological intervention and is delivered via brief face-to-face instruction followed by a mobile-app-supported self-directed practice. The scalability of this type of brief cognitive fitness training intervention may prove to be both attractive and viable in the area of workplace sustainability, including high-risk settings, such as the military and first responders. The high completion rate amongst our participants suggests that combining brief face-to-face instruction with a mobile-app-supported self-directed practice is likely to improve the user uptake for interventions of this type. On a broader note, our study confirmed that a combination of front-loaded face-to-face instruction with app-supported self-paced practice can produce new capability to deliver an effective mental health intervention on a large scale, and can improve its sustainability in the form of instructor/trainee ratio and enhanced acceptance by the end-user.

The effects of our multimodal cognitive fitness training program in improving mood, resilience, and burnout symptoms suggest that further investigation aimed at optimising the parameters of such an intervention is worthwhile. This could take the form of a larger-scale randomised controlled trial with a longer-term follow-up and the incorporation of measures of on-the-job performance in addition to self-reported mood, resilience, and burnout, to ascertain the impact on sustainable performance in high-pressure workplaces.

**Author Contributions:** Conceptualisation, P.T. and E.A.; methodology, P.T. and E.A.; software, P.T.; validation, E.A., F.R.W. and A.H.; formal analysis, P.T., E.A. and A.H.; investigation, P.T.; resources, P.T.; data curation, P.T.; writing—original draft preparation, P.T.; writing—review and editing, E.A., F.R.W. and E.A.; visualisation, P.T., E.A. and A.H.; supervision, E.A.; project administration, P.T. and E.A. All authors have read and agreed to the published version of the manuscript.

**Funding:** This research received no external funding.

**Institutional Review Board Statement:** The study was approved by the DEFENCE SCIENCE AND TECHNOLOGY GROUP'S Low-Risk Ethics Research Panel (LREP protocol LD 11-20, approved 30 July 2020) and was endorsed by the participants' chain of command.

**Informed Consent Statement:** Informed consent was obtained from all of the subjects involved in the study.

**Data Availability Statement:** The data presented in this study are available upon request from the corresponding author. The data are not publicly available due to privacy restrictions.

**Acknowledgments:** The authors dedicate special thanks to Paul Hannigan, for assistance with the project.

**Conflicts of Interest:** Taylor had a previous financial interest regarding the Resilient Mind Program, but Walker, Aidman, and Heathcote have no conflict of interest of relevance to the submission of this project.

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
