# Peer review of "Effects of Multimodal Physical and Cognitive Fitness Training on Sustaining Mental Health and Job Readiness in a Military Cohort"

_sustainability, doi:10.3390/su15119016_

Round 1
Reviewer 1 Report
Dear authors, thanks for submitting your manuscript. I enjoyed reading it, the research was clearly and concise ly presented.
1. There are two syntax errors which I detected: l. 79: include has (been criticized); l. 149: include to (the web)
2. l. 119: please explain the randomized design in more detail: what you mean by "work teams were allocated". How many groups were allocated with how many participants in each group?
3. lines 93 - 102 are repeated in lines 127 - 136, please try to shorten.
4. Figure 1 is too small and not readable. I suggest to put that figure into an online file which can be retrieved by readers.
5. the main limitation of the study is the lack of control group (l. 269). Indeed, this is a serious flaw. On the other hand, there is a remarkably low dropout rate which is an advantage of your study. If possible, please give some more information on former studies with the RMP program, regardless of the target group. Are there more results available about the effects of an RMP program and an FI program from other studies? If yes, this could strengthen your argumentation.
see my comments above
Author Response
Many thanks for your constructive input. We have actioned your points below as follows:
- The 2 syntax errors have been corrected.
- We have clarified the work teams point as follows
"In a block-randomized design, two work teams (who worked together on shifts, known as ‘watches’) of 40 and 38 participants were randomly allocated into one of two conditions;
- i) Standard Resilient Mind Program: three hours of face-to-face delivery plus web application for 4 weeks (RMP, n=38)
- We have removed the double-up of the description of the RMP.
- We will upload that image as a file, as suggested.
- We have added additional detail on the RMP and FI interventions in lines 106-121, as follows:
"The RMP has previously been shown to be effective in improving self-reported wellbeing and resilience in corporate workplaces, including banking, insurance and government sectors [24]. The current study also examined the additional effect of Functional Imagery practice (FI) on the outcomes of RMP. FI is an imagery-based behavioral intervention designed to help people enhance their motivation and sustain behavior through mental imagery. It typically combines person-centered counselling such as motivational interviewing, with tailored imagery exercises to elicit strong emotional responses and strengthen motivation for change [25,26]. FI has been shown to enhance grit in soccer players [25], and assist in weight loss [26] and the reduction alcohol cravings [27]."

Reviewer 2 Report
This study has strengths and weaknesses. First, it is professionally written and executed, and easy to read. The authors are upfront with potential weaknesses regarding the study, which is very positive. The main strength in the paper is the analysis of the Functionary Imagery practice. This because participation in this practice was block-randomized among the participants in the study as a whole. The results are interesting, as no effect was found for this "extra" practice. The weakness of the study is of course that overall, the participants were not randomly chosen, and the overall result that the program in total had positive effects may be due to selection bias. The authors are very clear on this which is a good thing.
Overall the study provides us with some new information regarding the Functionary Imagery practice. In the paper this fact could be emphasized bit more.
Author Response
Many thanks for your constructive feedback. We have added additional detail on the FI training in 2 sections:
1. Lines 115 to 121 contain more detail on FI practice;
"The current study also examined the additional effect of Functional Imagery practice (FI) on the outcomes of RMP. FI is an imagery-based behavioral intervention designed to help people enhance their motivation and sustain behavior through mental imagery. It typically combines person-centered counselling such as motivational interviewing, with tailored imagery exercises to elicit strong emotional responses and strengthen motivation for change [25,26]. FI has been shown to enhance grit in soccer players [25], and assist in weight loss [26] and the reduction alcohol cravings [27]."
2. In the discussion, we expanded on methodological factors that could have contributed to the lack of observable effect of the FI practice;
"There are, nevertheless, several methodological factors that may have contributed to the lack on any observable FI impact. First, and most importantly, our induction of FI practice may have been sub-optimal – participants in our study were given a brief guide to FI practice and instructed to use the feature in the app, whereas in other FI interventions, participants are guided through a series of mental imagery exercises and trained to practice imagery at home [24 – 26]. "
Hopefully this is to your satisfaction.

Reviewer 3 Report
I appreciate being able to review this article. I really enjoyed reading it. I congratulate the authors for the work. It is very well constructed, addresses a socially relevant topic. Methodologically it is well structured. However, I think it should make some changes before being approved for publication:
1. The structure of the article should follow the APA norms. In the section "Method" the first subsection should be "Research design". In this section you must identify the type of design and support it with relevant bibliography. The second subsection of "Method" should be "Participants" and the type of sampling performed should be described. The third subsection should be "Materials/Instruments". And the fourth subsection should be "Procedure". Please review some previously published articles.
2. The sample is open to criticism. They should justify that the sample size allows generalizable results. I advise you to perform a generalizability analysis.
3. They should calculate the effect size and incorporate it into the summary,
I will be watching for modifications.
Author Response
Many thanks for your most constructive input. Please see attached the amended document and below for our response to your points.
- We have edited the methods section as suggested. The only exception was that we kept 'Procedure' before 'Materials', as we felt if had a better logical flow. We will, however, change that if you think otherwise.
- We have added the following paragraph on generalizability under section 2.5 Analysis (lines 263 to 270): "Based on meta-analytic evidence from randomized-control studies of the effects of similar programs on stress [34], effect size of d = 0.4 on training-induced change can be considered meaningful and practically significant. This effect size is equivalent to f = 0.25 for ANOVA power calculations. To reliably detect effects of this size and to improve the generalizability of findings, with two groups and two repeat measurements in our design, G*Power calculations return a minimal sample size of 27 participants per group [35]. To allow for sample attrition, 40 and 38 participants were recruited to the two groups."
- We have added effect sizes on both the Abstract and the results section.

Round 2
Reviewer 3 Report
I believe that the authors have adequately answered the questions raised. However, they should modify the order of the structure:
The structure of the article should follow APA guidelines. In the section "Method" the first subsection should be "Research design". In this section you should identify the type of design and support it with relevant bibliography. The second subsection of "Method" should be "Participants" and the type of sampling performed should be described. The third subsection should be "Materials/Instruments." And the fourth subsection should be "Procedure". Please review some previously published articles.
Author Response
Hi there,
This has now been actioned and the updated word document is attached.
